# Influence of the Surface Material and Illumination upon the Performance of a Microelectrode/Electrolyte Interface in Optogenetics

**DOI:** 10.3390/mi12091061

**Published:** 2021-08-31

**Authors:** Junyu Shen, Yanyan Xu, Zhengwen Xiao, Yuebo Liu, Honghui Liu, Fengge Wang, Wanqing Yao, Zhaokun Yan, Minjie Zhang, Zhisheng Wu, Yang Liu, Sio Hang Pun, Tim C. Lei, Mang I Vai, Peng Un Mak, Changhao Chen, Baijun Zhang

**Affiliations:** 1School of Electronics and Information Technology, Sun Yat-sen University, Guangzhou 510275, China; shenjy7@mail2.sysu.edu.cn (J.S.); xuyy87@mail2.sysu.edu.cn (Y.X.); xiaozhw9@mail2.sysu.edu.cn (Z.X.); liuyueb@foxmail.com (Y.L.); liuhh36@mail2.sysu.edu.cn (H.L.); wangfg@mail2.sysu.edu.cn (F.W.); yaowq@mail2.sysu.edu.cn (W.Y.); yanchk@mail2.sysu.edu.cn (Z.Y.); zhangmj56@mail2.sysu.edu.cn (M.Z.); wuzhish@mail.sysu.edu.cn (Z.W.); liuy69@mail.sysu.edu.cn (Y.L.); 2State Key Laboratory of Optoelectronic Materials and Technologies, Sun Yat-sen University, Guangzhou 510275, China; 3State Key Laboratory of Analog and Mixed-Signal VLSI, Institute of Microelectronics, University of Macau, Macau 999078, China; lodgepun@um.edu.mo (S.H.P.); fstmiv@um.edu.mo (M.I.V.); chen.chang.hao1@gmail.com (C.C.); 4Department of Electrical Engineering, University of Colorado Denver, Denver, CO 80204, USA; tim.lei@ucdenver.edu; 5Department of Electrical and Computer Engineering, Faculty of Science and Technology, University of Macau, Macau 999078, China; fstpum@umac.mo

**Keywords:** optogenetics, optrode, microelectrode/electrolyte interface, surface material, photostability, optical artifact

## Abstract

Integrated optrodes for optogenetics have been becoming a significant tool in neuroscience through the combination of offering accurate stimulation to target cells and recording biological signals simultaneously. This makes it not just be widely used in neuroscience researches, but also have a great potential to be employed in future treatments in clinical neurological diseases. To optimize the integrated optrodes, this paper aimed to investigate the influence of surface material and illumination upon the performance of the microelectrode/electrolyte interface and build a corresponding evaluation system. In this work, an integrated planar optrode with a blue LED and microelectrodes was designed and fabricated. The charge transfer mechanism on the interface was theoretically modeled and experimentally verified. An evaluation system for assessing microelectrodes was also built up. Using this system, the proposed model of various biocompatible surface materials on microelectrodes was further investigated under different illumination conditions. The influence of illumination on the microelectrode/electrolyte interface was the cause of optical artifacts, which interfere the biological signal recording. It was found that surface materials had a great effect on the charge transfer capacity, electrical stability and recoverability, photostability, and especially optical artifacts. The metal with better charge transfer capacity and electrical stability is highly possible to have a better performance on the optical artifacts, regardless of its electrical recoverability and photostability under the illumination conditions of optogenetics. Among the five metals used in our investigation, iridium served as the best surface material for the proposed integrated optrodes. Thus, optimizing the surface material for optrodes could reduce optical interference, enhance the quality of the neural signal recording for optogenetics, and thus help to advance the research in neuroscience.

## 1. Introduction

In recent years, optogenetics and opto-electrophysiology have emerged as an interdisciplinary research topic combining neuroscience, optics, semiconductor optoelectronics and bioengineering. The optrode, a kind of compact device, appeared with the growth of optogenetics and has been becoming a significant tool in neuroscience through the combination of offering accurate spatial and temporal stimulation to target cells and recording biological signals simultaneously during experiments [1]. Optrodes can utilize light to replace electricity as the stimulating tool to bring the millisecond-scale precise and harmless stimulation. In addition, they can be easily integrated with massive parallel neural recording microelectrodes to accurately detect the potentials of neurons simultaneously [2]. Compared with traditional electrical stimulation, the application of optrodes in optogenetics has lots of advantages, such as a high spatial and temporal resolution, high consistency of stimulation and recording, excellent biocompatibility with little tissue damage, reversible regulation, miniaturization and intelligent control [1,2,3].

Clinically, optogenetics is opening up new neuromodulation therapeutic approaches to treat or to control neurological diseases [4]. Initial results indicate that optogenetic therapy is effective to partially restore vision in photoreceptor-deficient mice. Currently, there are two ongoing FDA-approved clinical trials using optogenetic opsins to treat Retinitis Pigmentosa (RP)—a group of genetic disorders causing a progressive loss of photoreceptors and resulting in difficulty to see at night and loss of peripheral vision. The two trials transfect Channelrhodopsin-2 (ChR2) DNA to retinal ganglion cells of RP patients to attempt to restore their vision [4]. Recently, J. Sahel combined the intraocular injection of an adeno-associated viral (AAV) vector encoding ChrimsonR with light stimulation via engineered goggles for treating RP. It was the first reported case of partial functional recovery in a neurodegenerative disease after optogenetic therapy [5]. Therefore, the development of the integrated optrodes not only provides a new tool for neuroscience research, but also has a great potential to be used in future treatments of clinical neurological diseases like Parkinson’s disease, depression and epileptic seizures, and even for controlling the activity downstream from sites of damage or degradation of the nervous system [2,6,7,8,9]. It can be anticipated that clinical trials using optogenetics for treating brain diseases in human patients should not be far too distant in the future.

To date, optrodes can be classified roughly into four types. The earliest optrodes were fiber-based, assembling huge glass fibers with recording probes [10,11,12]. Then, electrocorticography (ECoG) optrodes were followed but mainly focused on neurons on the surface of brains or skulls [13,14]. Utah optrodes can easily cause large-area tissue damage [15,16,17,18,19]. The current state-of-art monolithic integrated planar optrodes are called Michigan optrodes, first designed by researchers from the University of Michigan, Ann Arbor. Michigan optrodes integrate planar microelectrodes with implantable light sources like micron-scale laser diodes or LEDs. The integrated planar optrode achieves a high integration level by using a precise fabrication process. Compared with other optrodes, this integrated optrode is fiber-free and able to minimize the probe volume, maximize the integration level and achieve a multi-functional layout. Instead of using an external light source, the fiber-free solution is more convenient for behavior experiments on animals without limiting their activities. Thanks to the higher integration level, the optrodes can be minimized to a small needle size for a lower implant insertion damage and a better biocompatibility [20,21,22]. Other than that, more micro-LEDs and microelectrodes are integrated in the optrode to achieve a higher spatial resolution both in biological signal recording and optical stimulation [23,24]. In addition, a shielding layer can be integrated into the optrode to alleviate the electrical artifacts both from the light sources and the power-line. However, an integrated planar optrode is still suffering from the electrical artifacts and optical artifacts. Optical devices like LEDs induce electromagnetic interference from electrical driving signals, and it is easy to introduce electrical artifacts to the biological signal recording [25]. Other than that, it has been reported that the optical artifacts can be found on the microelectrodes, when illumination is introduced to the microelectrode/ electrolyte interface. Optical artifacts are the inevitable problem of all types of optrodes, and the explanations for this artifact have yet to reach an agreement [26,27,28,29,30,31].

Nowadays, it is critical for optrodes to be optimized, in order to enhance the quality of the recorded signals and improve the reliability during long-term neural implants. To illuminate a large enough brain region, a high optical power is usually required because of the exponential attenuation of light in the brain, leading to the accompanied large optical artifacts in the neural recordings [32]. Therefore, the strategies for eliminating optical artifacts and improving the neural signal are being investigated around the world. There are several speculated physical mechanisms for explaining the optical artifacts, such as the photovoltaic effect [27,33], photoelectrochemical effect [27,34] and photothermal effect [27], which have yet to reach an agreement. Strategies for avoiding optical artifacts can be mainly divided into two types. One of them is through the modification of microelectrodes, like electroplating gold or platinum electrodes with pulsed current for a larger contact area [35,36] and modifying metal electrodes with conductive polymers [31,37,38]. The other one is to use transparent microelectrode materials, such as indium tin oxide or graphene, both of which have been proved to be able to reduce the optical artifacts effectively [33,39].

In this work, we propose a new generation of monolithic integrated planar optrode, which was based on Gallium Nitride (GaN) on sapphire instead of silicon. To optimize this optrode, we propose a model to explain the charge transfer mechanism on the microelectrode/electrolyte interface based on this device. A method to measure the electrical properties of the optrodes under different conditions was also proposed to verify the accuracy of the model and investigate the influence of light, which few studies had reported in detail before. Moreover, microelectrodes on the optrodes were optimized with different surface materials, and a corresponding evaluation system was further built to assess the performances of optrodes with different surface materials. 

It was noteworthy that photostability and optical artifacts were measured and added to the evaluation system through the investigation of the optical influence upon the interface. According to our measurement, iridium could be considered as the best surface material for microelectrodes on optrodes among five selected metals, which were biocompatible and had all been used before for neural recordings [40]. The optimized optrode was expected to obtain more precise stimulated results in optogenetic control and could be potentially applied in the clinical treatments of neurological diseases.

## 2. Experiments and Methods

### 2.1. Design and Fabrication

The proposed sapphire-based integrated planar optrodes, which could be a good alternative to the silicon-based ones to become the next generation optrodes, were integrated by multiple electrophysiology channels and a high intensity micro-LED. The sapphire-based micro-LEDs could achieve a high optical density, that was bright enough for optogenetic protein activation. In addition, the high material stiffness of sapphire made the optrodes much more robust and more suitable for long-term neural implants. Moreover, sapphire is optically transparent, for the light to pass through the optrode itself, which allowed a 4π steradian optical illumination coverage for an effective optogenetic stimulation of the target brain region.

Figure 1a,b illustrates the top and cross-sectional view of the proposed integrated planar optrode. The optrode consisted of four recording microelectrodes with a diameter of 10 μm, which are in the same scale of our target neurons in the biological experiments, and a GaN/InGaN multiple quantum-wells (MQWs) blue micro-LED as an excitation light source. The overall dimensions of the optrodes were minimized to alleviate tissue damage, while a sufficient mechanical strength for insertion was maintained. The length of the optrode was designed to be 12-mm-long, to allow for a sufficient insertion in futural biological experiments, and the width could not be less than 0.3 mm because of the limitation of wafer-sawing and wire-bonding. The GaN/InGaN micro-LED used in our fabrication was epitaxially grown on a sapphire substrate using metal organic chemical vapor deposition (MOCVD) system, comprising a GaN buffer layer, an *n*-GaN layer, GaN/InGaN MQWs, an AlGaN electron blocking layer and a *p*-GaN layer, from the bottom to the top.

The detailed fabrication steps were as follows. Firstly, a 400-nm-depth mesa isolation was etched to an *n*-GaN layer by the inductively coupled plasma (ICP). Then, a *p*-GaN layer was activated at 830 °C for 30 min in an alloy furnace. Secondly, a metal layer of Ni/Au (5/10 nm) was deposited on the mesa as a transparent conductive layer (TCL) and followed by rapid thermal annealing for 180 s with ambient temperature at 600 ℃ to form a *p*-type electrode. Thirdly, a 300-nm-thick SiO_2_ layer was deposited by the plasma-enhanced chemical vapor deposition (PECVD) to form an isolation layer, in order to protect the device from short-circuiting when fabricating long interconnections and bonding pads. Then, metal layers of Ti/Au (50/100 nm) were deposited to create the cathode and anode for blue micro-LEDs, as well as interconnection metal lines and bonding pads. Afterwards, different electrode materials were deposited on microelectrode sites for further investigation, including a 20-nm-thick nickel layer for adhesion and a 35-nm-thick surface material. Five kinds of surface material were utilized in our experiment, namely titanium (Ti), chromium (Cr), gold (Au), platinum (Pt), and iridium (Ir). All fabrication parameters were equally controlled to ensure the quality of the metals. The materials were all smooth on the surface, with the same real open areas contacting electrolyte. The detailed surface roughness and AFM micrographs are shown in Figure A1 in Appendix A. Lastly, another SiO_2_ layer was deposited as the passivation layer, and the windows for microelectrode recording sites and bonding pads were opened by the chemical etching. The encapsulation and assembly were followed by the fabrication process. Probes were acquired from the wafer through back-grinding and wafer-sawing. Then the probe was fixed on the customized printed circuit board (PCB) with glue. The p/n electrode pads of LED and the microelectrode pads on the probe were connected to the corresponding electrode pads on PCB by a wire-bonding technique, as shown in Figure 1c,e, so that the micro-LED on the probe could be conveniently driven by the external current, and the neural signals which were recorded by the microelectrodes could also be extracted for further investigation. The epoxy was finally applied to protect the wire-bonded area from short-circuiting. During the implantation surgery, the PCB board was mounted on a micro-drive of a stereotaxic device, and the tip of the optrode was slowly driven into the brain to reach the target area. The finished optrode is shown in Figure 1c. Figure 1d shows the micrograph of the top area of the optrode, which was designed to be implanted into neural tissues, and Figure 1e shows the micrograph of the wire-bonded area protected by epoxy.

### 2.2. Theoretical Model and Equivalent Circuit

The microelectrode/electrolyte interface plays an important role when the optrode records electrophysiological signals from the target neurons. The signal forms when charges in the electrolyte and microelectrodes transfer to each other through the interface. Figure 2 illustrates the theoretical model and equivalent circuit to describe the charge behaviors on the interface. As shown in Figure 2a, two types of charge transfer processes occur on the surface of microelectrodes in electrolyte, namely electrical double layer (EDL) capacitor and faradaic processes.

*EDL capacitor:* With a given limited potential in a small range, little charges can transfer across the microelectrode/electrolyte interface, and the charges and oriented dipoles accumulate on the interface, behaving like a capacitor. The Gouy–Chapman-Stern (GCS) model modified by O. Stern explains this capacitor behavior. As shown in the lower part of Figure 2a, the electrolyte side of the capacitor is thought to be made up of two layers. The layer closest to the microelectrode is the inner Helmholtz plane (IHP) at x_1_, which contains solvent molecules and other specifically adsorbed species. The second layer is called outer Helmholtz plane (OHP), where solvated cations accumulate because of long-range electrostatic force. The locus of the electrical center of OHP is x_2_, which is more distant from the metal surface due to the finite size of the solvated cations. Hence, the interface forms an EDL capacitor with IHP and OHP, and occurs because of the charge-discharge processes of the capacitor [41].

*Faradaic processes:* As shown in the upper part of Figure 2a, charges transfer across the microelectrode/electrolyte interface to form currents because of reduction and oxidation (redox) reactions. Faradaic processes include mass transfer, electron transfer, chemical reaction preceding or following electron transfer and other surface reactions like adsorption and desorption [41].

As shown in Figure 2b, a parallel structure is introduced as the equivalent circuit to model the charge transfer mechanism, as the total current across the microelectrode/electrolyte interface is the sum of faradaic processes and charge–discharge processes of the EDL capacitor. C_d_ represents the section of EDL capacitor, which is nearly a pure capacitor. The elements simulating faradaic processes are separated to charge transfer resistance R_ct_ and Warburg impedance Z_w_, which represent mass transfer and diffusion. R_Ω_ represents the resistance of the bulk solution. The impedance of the microelectrode/electrolyte interface (Z_M/E_) can be described as Equation (1) [41].
(1)$$Z_{M/E}=R_{\Omega }+\frac{1}{{j\omega C}_{d}+\frac{1}{R_{\mathrm{ct}}+Z_{W}}}$$

### 2.3. Testing System

As shown in Figure 3a, the measurement of the electrical properties of the microelectrode/electrolyte interface, like impedance and capacitance, were carried out by an LCR meter (Keysight E4980A) and a semiconductor device analyzer (Agilent B1500A), respectively, at room temperature. During the measurement, the top of the optrode was immersed into a phosphate buffer solution (PBS) (0.9%, pH ≈ 7.1~7.2), which was similar to the body fluid of mammals, so that the neural tissue could be simulated in the measurement. This testing environment was similar to a two-electrode cell in electrochemistry. In PBS, microelectrodes played the role of the working electrode and a tungsten probe served as the reference electrode. In this testing system, the amplitude of the applied testing AC signal (V_Level_) was fixed at 50 mV, while the reference bias (V_Ref_, bias voltage between microelectrodes and tungsten probe), frequency, surface materials and light conditions were variables for investigating the performances of microelectrodes.

The ability to avoid the optical artifact is another important characteristic for optrodes. As shown in Figure 3b, the testing system for measuring optical artifacts was composed of the optrode, light source, amplifier [42], data acquisition card (National Instrument NI USB-6215) and shielding setup. The top of the encapsulated optrode, which included a micro-LED and microelectrodes, was immersed into the grounded PBS. The bonding pads of the microelectrodes were connected with the amplifier, and the output of the amplifier was sent to the data acquisition card to acquire optical artifacts. Both integrated micron-scale LEDs and external light sources provide pulsed light to the optrodes to measure optical artifacts which could also be recorded in this testing system. Taking the EMI interference of the light sources into consideration, the shielding setup was utilized to protect the microelectrode/electrolyte interface from external interference when external light sources were used.

## 3. Results and Discussion

### 3.1. Electrical Performance Characterization

It is well known that the impedance of a microelectrode/electrolyte interface (Z_M/E_) is one of the most important electrical characteristics of the microelectrode, which reflects the charge transfer capability, recording performance and signal quality for neural recording. Impedance measurement is used to apply an AC signal in a small amplitude to the microelectrode/electrolyte interface and to observe the responses at a steady state [31]. Figure 4 shows the electrochemical impedance spectroscopy (EIS) of the microelectrodes, illustrating the variation of Z_M/E_ versus frequency in a logarithmic scale. Microelectrodes of different surface materials with a diameter of 10 μm were measured under the same testing environment, in which V_Level_ was 50 mV and V_Ref_ was 0, while the illumination was not added. In general, Z_M/E_ declined exponentially with the increase of frequency. Among these microelectrodes, the ones made of Pt and Ir showed the best electrical performance, followed by Au and Ti, and the worst metal material was Cr. In particular, a Z_M/E_ around 1 kHz is significant for the optrode, because the pulse width of the neural action potential is around 1 ms. Z_M/E_ values of Pt and Ir microelectrodes were measured to be 4.77 MΩ and 4.80 MΩ, respectively, at 1 kHz, much lower than those of Au (10.96 MΩ), Ti (13.39 MΩ) and Cr (16.52 MΩ).

According to the theoretical model in Section 2.2, it can be known that the Z_M/E_ value depends on the charge transfer on the microelectrode/electrolyte interface made by different surface materials. In order to further verify the influences of C_d_ and faradaic processes on Z_M/E_, the testing environment (two-electrode cell) could be considered as a “black box”, in which a certain excitation was applied and a certain response was measured.

The EDL capacitor C_d_ could be considered as a pure capacitance on the microelectrode/electrolyte interface, and be measured with the applied AC signal, while the V_Level_ was 50 mV at 1 kHz and V_Ref_ changed within a small range, from −50 mV to 50 mV. The insert of Figure 5a demonstrates the changing curve of C_d_ versus V_Ref_. According to our measurement, C_d_ stayed constant within the small range of V_Ref_ for every specific surface material, but increased successively from Cr, Ti, Au and Ir to Pt. Z_M/E_ and C_d_ values at V_Ref_ = 0 around 1 kHz were extracted and the Z_M/E_–C_d_ curve was fitted based on Equation (1). As shown in Figure 5a, the change of Z_M/E_ versus C_d_ coincided with Equation (1), with the hypothesis that faradaic processes were neglected when the small amplitude neural signals were applied to the microelectrodes, demonstrating that C_d_ was the leading factor determining Z_M/E_. C_d_ was verified to exist on the microelectrode/electrolyte interface and transfer charges or electrons through charge–discharge processes, and the measurements could be repeated.

A few bubbles were observed on the microelectrode surface when V_Level_ or V_Ref_ reached 1 V or higher, which verified the generation of redox reactions. However, the faradaic processes were difficult to characterize due to their complexity, and were too weak to be observed at relatively low V_Level_ or V_Ref_. In this work, the Z_M/E_–V_Ref_ curves of microelectrodes within a V_Ref_ ranging from 0 to 500 mV were measured to characterize the faradaic processes. As shown in Figure 5b, Z_M/E_ values had similar attenuating trends versus V_Ref_ for every surface material. Z_M/E_ values began to decline rapidly, showing that the quantities of transferring charges from faradaic processes became larger and larger, and the measurement could not be repeated because microelectrodes were damaged by the intense redox reactions. It was speculated from experimental results that the faradaic processes gradually became the leading factor in determining charge transfers with the reference bias growing. Furthermore, a critical value (V_c_) was defined as the V_Ref_ value corresponding to the moment when Z_M/E_ dropped by 5% of the initial value, and the whole process from 0 to 500 mV was separated into two parts. The first part was called the steady stage, when V_Ref_ had not reached the critical value. In the steady stage, Z_M/E_ stayed almost constant or descended extremely slowly with the rise of V_Ref_, showing that the faradaic processes were weak. The second part was the intense stage, when V_Ref_ exceeded V_c_, in which Z_M/E_ decreased sharply with the increase of V_Ref_ and faradaic processes replaced C_d_ as the leading factor affecting charge transferring.

The effects of the surface material on C_d_ and the faradaic processes were investigated further for microelectrode evaluation. As shown in Figure 5a, Z_M/E_ declined as C_d_ rose in the sequence of Cr, Ti, Au, Ir and Pt. Choices of surface materials could improve the solid–liquid interfacial tension of the microelectrode/electrolyte interface, and lead to a better the EDL capacitor C_d_ [41]. A larger C_d_ could reduce the impedance of the microelectrode effectively, and a lower Z_M/E_ allows for a larger quantity of charges to transfer faster, leading to a higher signal-to-noise ratio of the recorded neural signals. According to our measurement results, Pt and Ir had the best charge transfer capacity among the group, which could reduce the electrical noises and enhance the quality of the recorded signals.

In Figure 5b, Z_M/E_–V_Ref_ curves were analyzed to further evaluate the electrical stability of microelectrodes. The electrical stability for one material was defined as the difficulty degree of occurrence intense redox reaction in faradaic processes. Some surface materials with higher electronegativities were more attractive to electrons, and possessed a better electrical stability for faradaic processes. To distinguish the steady and intense stages and represent the difficulty degree of occurrence of a redox reaction, a critical value (V_c_) was defined in this work. The critical values and electronegativities of the five surface materials were extracted and compared in Table 1. The three referred sets of electronegativity scales were the Pauling value (X_p_) [43], the Allred–Rochow value (X_a_) [44] and another set of electronegativities (X_Y_) of elements of valence states which were calculated on the basis of the electrostatic force [45]. In general, a material with higher electronegativity had a larger critical value, showing that these materials are more attractive to electrons and less able to create redox currents, so that they have a better electrical stability. It is worth mentioning that Ir microelectrodes stood out for their large critical value and excellent electrical stability. The reason for this phenomenon might be due to the fabrication processes of Ir microelectrodes in a high-power electron-beam, which would produce a high temperature and make some iridium oxide form [14,46] during the fabrication. The slopes of the Z_M/E_–V_Ref_ curves in intense stages indicated the intensity of the redox reactions for different surface materials in Figure 5b, and Ir microelectrodes also had the mildest reaction intensity degree. It could be concluded, from the measurement results, that Ir had the best electrical stability among the group.

Figure 5c compares the Z_M/E_ values before and after the redox reaction under V_Ref_ = 500 mV, while the measurement condition was kept constant (V_Ref_ = 0 and V_Level_ = 50 mV at 1 kHz). The Z_M/E_ values of Cr and Au microelectrodes had a dramatic drop, while the Ti and Ir ones remained relatively stable and the Pt ones performed best, leading to the finding that the microelectrodes of Ti, Ir and Pt had better electrical recoverability. The electrical recoverability was defined as the change of Z_M/E_ values before and after the intense redox reaction, which can happen when the electrode was used for electrical stimulation or radio frequency ablation. Electrical recoverability could be utilized to assess long-term reliability and the lifetime of microelectrodes after they were putted into neural implants. Besides, the surface materials used for evaluation had all been shown to have a good biocompatibility, and a good performance on neural recording [14,28,40,47,48]. It was reported that the recording duration of Ir electrodes could be up to 72 weeks, slightly longer than that of Au and Pt electrodes (52 weeks) [40].

### 3.2. Photostability and Optical Artifacts Characterization

With the increase of the integration degree of the optrodes, the illumination can induce optical artifacts to interfere with the microelectrode/electrolyte interface and contaminate the recorded neural signals [26,27,28,29,30,31]. In order to get a higher signal-to-noise ratio for the measured neural signal, eliminating optical artifacts becomes an important function for optrodes [29,33,49], so it is necessary to investigate the influence of light upon the charge transfer on the interface. Photostability was introduced to represent the stability of the interfaces made by different surface materials under illumination for further comparison. Optical artifacts were also measured to investigate the relationship between optical artifacts and various parameters to evaluate the performance of the optrodes

To investigate photostability, the way in which illumination affected the microelectrode/electrolyte mechanism on the interface was studied through changing light conditions in the testing environment. The electrical signal driving the micro-LED on the integrated optrode might generate additional electrical noises like EMI, which interfered with the investigation of the optical influence. Therefore, external light sources were used in our experiments for changing the illumination conditions (wavelength and power) conveniently. Shielding was also applied to remove EMI and other electrical contaminations from various electrical noise sources in the laboratories. In this way, optical artifacts were distinguished from mixed noises for further investigation. 

The influence of illumination on the EDL capacitor C_d_, which was the vital component of the microelectrode/electrolyte interface, is demonstrated in Figure 6. Under the condition of V_Ref_ = 0 and V_Level_ = 50 mV at 1 kHz, the C_d_ values of various surface materials were measured under different light conditions, including darkness and illumination with wavelengths of 620 nm, 460 nm and 280 nm, respectively. The optical power density of different illumination wavelengths was controlled to be around 5.12 mW/mm^2^. When changing the light conditions from darkness to an illumination of 620 nm, the C_d_ value showed a rising trend for every surface material. According to the quantum theory of light [27], photons could provide energy to charges in electrolyte, which promoted their accumulation and polarization on the interface and increased the charge density of OHP and IHP. According to the measurement, C_d_ kept rising when the illumination wavelength dropped from 620 nm to 280 nm because photons with a shorter wavelength provided greater energy. From the increase of C_d_, it was concluded that the illumination could accelerate the accumulation of charges and cause charge–discharge processes of C_d_, resulting in part of the optical artifacts.

Based on the analysis of the optical influence on the microelectrode/electrolyte interface, the photostability of different surface materials could be further discussed. As shown in Figure 6, the C_d_ of Cr and Au microelectrodes grew more rapidly than other three materials. This indicated that fewer charges or ions accumulate on microelectrodes made by Ti, Pt and Ir, which maintained better photostability.

Figure 7 shows the applied optical signals in 463 nm and the corresponding measured optical artifacts of the optrodes with different surface materials. Figure 7a shows the typical pulsed optical signal used in our testing system. The optical signal was expressed in terms of voltage, and it was extracted from photocurrents detected by a photodiode connected with a 1 GΩ resistance in series. The optical power density of the illumination was controlled to be 16.8 mW/mm^2^ by a power meter (Newport 843-R) with a 1-cm-diameter detection area. This power density was higher than the reported minimum optical power density required, which was 10 mW/mm^2^ [27], to activate ChR2 in optogenetic control. Figure 7b–f shows the recorded optical artifacts of optrodes with different surface materials, measured by the setup showed in Figure 3b, with pulsed light. The pattern of the artifacts had a negative peak at the rising edge of the applied optical signal and a positive peak at the falling edge, and the positive and negative peaks both decayed to baseline in a short duration. As shown in Figure 7a, the rising time of the optical signal was much shorter than the falling time. Since there was a high pass filter, which was normally used to reject the noise in the low-frequency domain, in the neural recording amplifier that was used to record the optical artifacts, the artifact amplitude at the falling edge was smaller than the one at the rising edge.

Through the comparison of Figure 7b–f, the performance of the optrode in the optical artifacts was becoming better and better as the surface materials changed according to the specific order of Cr, Ti, Au, Pt and Ir. It was found that light could induce the most serious optical artifacts, which were 1.201 mV, as shown in Figure 7c, on optrodes made by Cr among all selected materials. In neural recordings, the extracellular action potentials generated by a neuron typically had an amplitude ranging between 50 to 500 μVpp with a rising time around 0.2 ms and a pulse duration around 1 ms [50,51]. This caused the neural signals to be easily affected, distorted or submerged in such large optical artifacts. Ti microelectrodes were also found to have relatively large optical artifacts, though they possessed good photostability in C_d_. By contrast, the transient amplitudes of the recorded optical artifacts of Ir and Pt optrodes were the lowest. Compared with Cr optrodes, the negative and positive peaks of the recorded optical artifacts from the Ir and Pt optrodes (Figure 7e,f) decreased by 46.8% and 44.7%, respectively, and were found to be highly correlative with the fact that Ir and Pt optrodes had the best performance in various properties, like the impedance discussed above. 

To further investigate the relationship between optical artifacts and other parameters of the optrodes, a Pearson correlation coefficients [52] of the recorded optical artifacts and other measurement results are shown in Table 2 and Table 3. The results of the relationships of the optical artifacts with Z_M/E_, V_c_, C_d_ and C_d_ growth rates are shown in Figure 8. According to the correlation analysis, the amplitudes of the recorded optical artifact were found to be significantly correlated (|r| = 0.98977 @ darkness, |r| = 0.98919 @ 620 nm, |r| = 0.99148 @ 460 nm, |r| = 0.99445 @ 280 nm) with C_d_ (electrical double-layer capacitor) under all the testing illumination conditions, and also significantly correlated (|r| = 0.97589) with Z_M/E_ (the impedance of microelectrode/electrolyte interface), as shown in Figure 8a,c. This phenomenon was possibly due to the fact that a lower Z_M/E_ or a higher C_d_ on the microelectrode/electrolyte interface allowed for a larger quantity of charges to transfer faster, and these charges formed a more stable microelectrode/electrolyte interface to oppose the disturb from illumination, leading to the fact that microelectrodes with a lower Z_M/E_ and a higher C_d_ could inhibit the optical inference better and get a lower artifact amplitude. Moreover, the recorded optical artifact amplitude was found to be moderately correlated (|r| = 0.72762) with V_c_, which reflected the electrical stability from the faradaic processes in Figure 8b. The microelectrode which had a higher V_c_ was more stable in the faradaic processes, and was probably more stable when the illumination was applied to the recording surface. In addition, the optical artifact amplitudes were poorly correlated with electrical recoverability after the redox reaction (|r| = 0.36151) and the photostability from the C_d_ growth rate under the illumination conditions of optogenetics (|r| = 0.48589 @ 460 nm). This indicates that the optical artifact is more related to Z_M/E_ and C_d_, and not to the changing rate of these two parameters under optical illumination.

As shown in Figure 8a,c, the amplitudes of the optical artifacts were proportional with the Z_M/E_ and C_d_ of the electrode, which were key parameters for the charge transfer capacity. This indicated that the optical artifacts of the optrode could be reduced by selecting suitable surface material for the microelectrodes. The metal with a better charge transfer capacity and electrical stability had a better performance on the optical artifacts, regardless of its electrical recoverability and photostability under the illumination conditions of optogenetics. It could be concluded that the metal with a lower electrode impedance, a higher EDL capacitor and a higher critical value would have a better performance on the optical artifacts. 

According to our results, Ir was the best surface material for the optrodes, with advantages for enhancing the recorded signal qualities and reducing the optical artifacts. Furthermore, it could be concluded that the influence of light upon the interface characteristics like Z_M/E_ or C_d_ could be the major cause of the optical artifacts. It was found that the optical illumination could lead to the changes of the EDL capacitor (C_d_) of the microelectrode/electrolyte interface, and this sudden change of the EDL capacitance could lead to charge–discharge processes on the rising and falling edge of the applied optical signal, resulting in high-amplitude optical artifacts in the recorded neural signals.

## 4. Conclusions

In this work, a robust integrated optrode, which had been integrated with multiple neural recording channels and a high intensity micro-LED on a sapphire substrate, was proposed. As a light-emitting material, GaN on sapphire could allow micro-LEDs to be fabricated directly on the substrates, which would simplify the encapsulation steps and enhance the efficiency of production. In addition, the sapphire substrate was less fragile and had a higher material stiffness than silicon, so sapphire-based optrodes was more durable for the implantation. Moreover, the transmittance of sapphire on visible band was high enough for micro-LEDs to illuminate the target brain region effectively.

To optimize the performance of this integrated optrode, the influences of surface materials and illumination upon the performance of the microelectrode/electrolyte interface were studied thoroughly in this work. It was found that Z_M/E_ values were related with the charge transfer on the interface, which consisted of the EDL capacitor and faradaic processes through a theoretical model analysis and experimental verification. More importantly, the influence of illumination on the EDL capacitor of the interface was found to be the possible major cause of optical artifacts’ occurrence. Furthermore, an evaluation system for optrodes with different surface materials was built up with the method of measuring Z_M/E_, C_d_ and optical artifacts under various testing environments, which varied in frequency, V_Ref_ and illumination conditions. In the evaluation system, the performances of the optrodes’ charge transfer capacity, electrical stability and recoverability, photostability and optical artifacts were comprehensively analyzed, and these characterizations could be optimized with a better selection of surface materials. Our results indicated that the metal with a better charge transfer capacity and electrical stability had a better performance on the optical artifacts, regardless of its electrical recoverability and photostability under the illumination conditions of optogenetics. Among all the selected metals, iridium had the best performance and served as the most suitable surface material for microelectrodes in optrodes, due to its lowest electrode impedance and optical artifacts, while platinum followed as the second best. All results pointed to the fact that iridium was expected to enhance the signal quality and reduce optical artifacts, which would greatly contribute to future works in optogenetics.

Although the proposed integrated planar optrode demonstrated a great potential to be used in optogenetics for its higher integration level in this work, in vitro, ex vitro and in vivo experiments are still needed to further verify optrodes in the more complex biological environments. Compared with the brain, the PBS solution could be oversimplified, while various biologically relevant, brain-present and redox active molecules (Cu (I/II), Fe (II/III) etc.) are missing, not to mention the nervous tissue response and inflammatory events that result in the failure of the implant electrode over extended periods of time. To overcome the limitations of the long-term use optogenetics in the brain, new design strategies and new materials with superior biocompatibility and artifact-free microelectrodes are currently being investigated.

## Figures and Tables

**Figure 1 micromachines-12-01061-f001:**
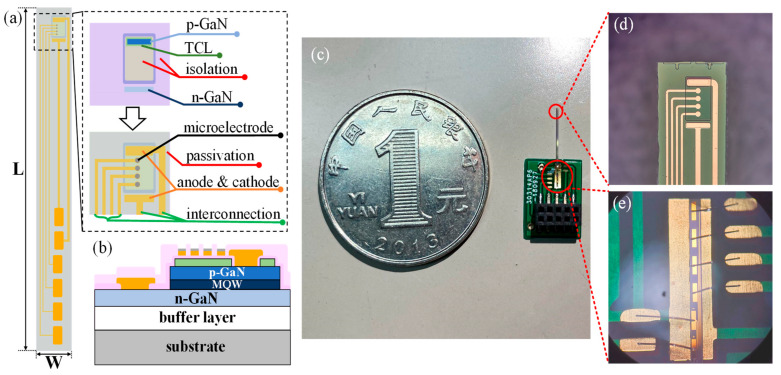
(**a**) Top view and (**b**) cross-sectional view of the monolithic integrated planar optrode. (**c**) The picture of the proposed encapsulated integrated planar optrode. (**d**) The micrograph of the top area and (**e**) the micrograph of the wire-bonded area on the optrode.

**Figure 2 micromachines-12-01061-f002:**
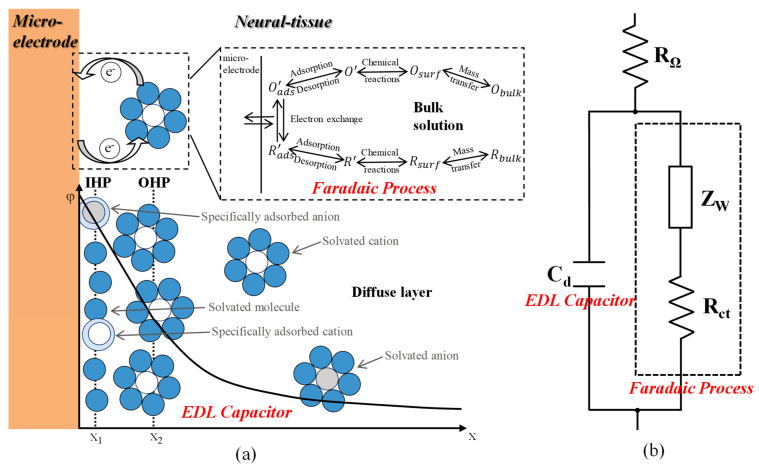
(**a**) Theoretical model for the charge transfer mechanism on the microelectrode/electrolyte interface, in which the upper part displays faradaic processes and the lower part an EDL capacitor. (**b**) Equivalent circuit for the charge transfer mechanism on the interface.

**Figure 3 micromachines-12-01061-f003:**
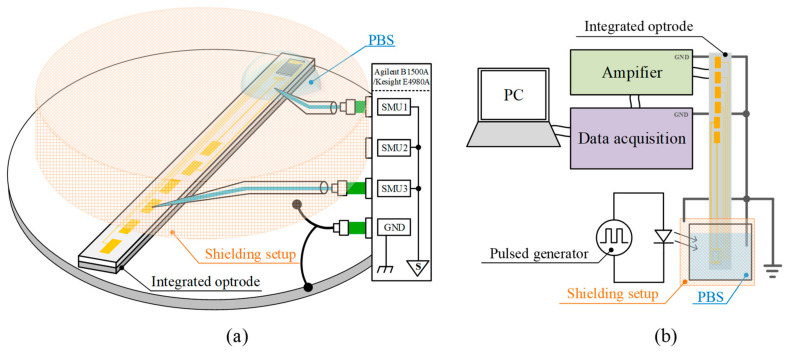
(**a**) The proposed testing system for electrical characterizations with a semiconductor device analyzer or LCR meter in PBS with shielding. (**b**) The testing system for optical artifacts with amplifier and data acquisition card in PBS with shielding, while the optical artifacts are generated by a light-emitting diode driven by a pulse generator.

**Figure 4 micromachines-12-01061-f004:**
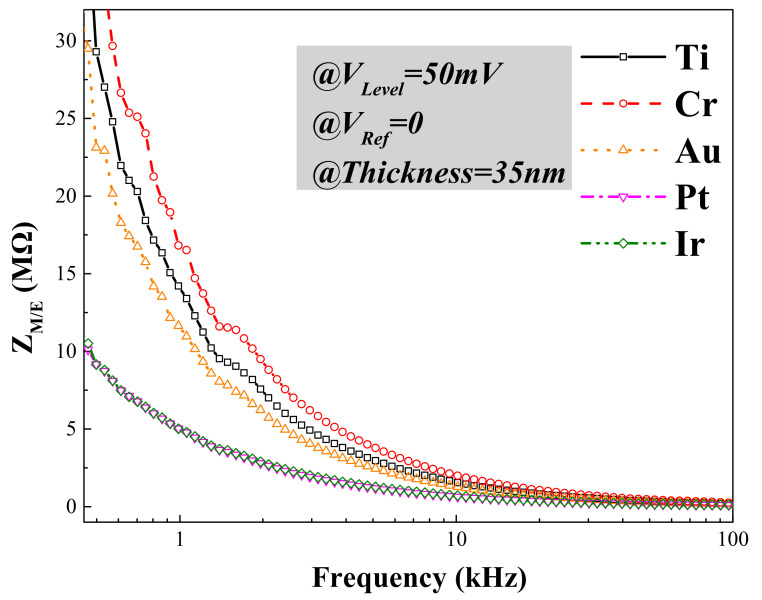
Electrochemical impedance spectroscopy (EIS) of microelectrodes with different surface materials, including Ti, Cr, Au, Pt and Ir.

**Figure 5 micromachines-12-01061-f005:**
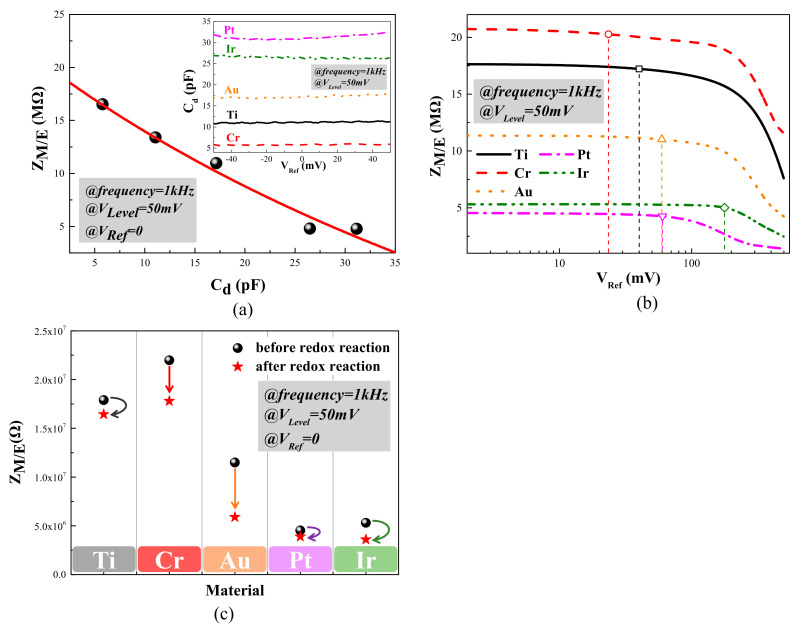
(**a**) Experimental values and fitting result of Z_M/E_ versus C_d_ (V_Ref_ = 0 and V_Level_ = 50 mV at 1 kHz). The insert is the C_d_-V_Ref_ curve of microelectrodes with different surface materials measured within the range of −50 to 50 mV, representing the EDL capacitor. (**b**) Z_M/E_-V_Ref_ curves of microelectrodes with different surface materials measured within the range of 0 to 500 mV, characterizing the critical values and electrical stability of faradaic processes. (**c**) Z_M/E_ values of microelectrodes with different surface materials measured before and after the faradaic processes (V_Ref_ = 0 and V_Level_ = 50 mV at 1 kHz).

**Figure 6 micromachines-12-01061-f006:**
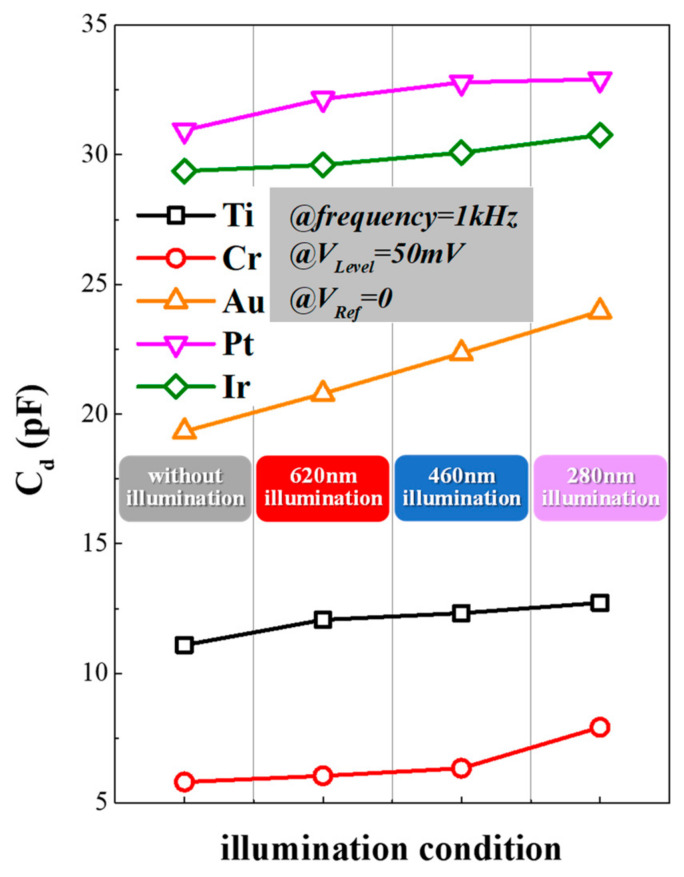
The change of C_d_ versus the wavelength (V_Ref_ = 0 and V_Level_ = 50 mV at 1 kHz) of microelectrodes caused by different surface materials.

**Figure 7 micromachines-12-01061-f007:**
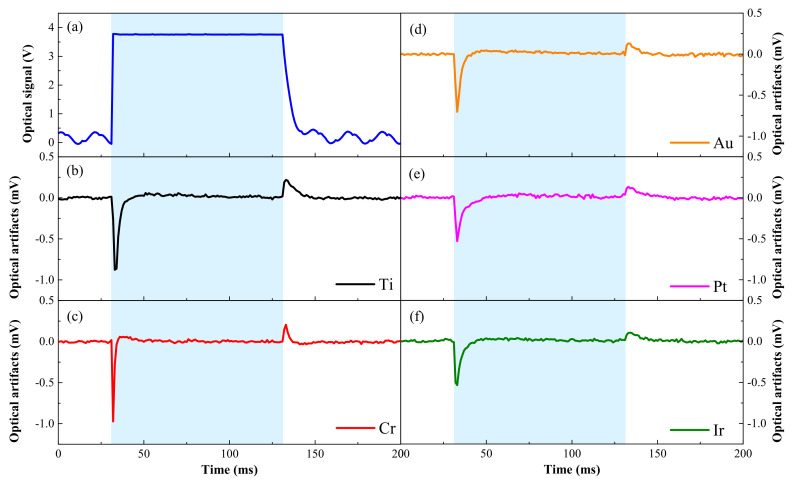
(**a**) The optical signal used in the testing system and the recorded optical artifacts of optrodes made by different surface materials, including (**b**) Ti, (**c**) Cr, (**d**) Au, (**e**) Pt, (**f**) Ir.

**Figure 8 micromachines-12-01061-f008:**
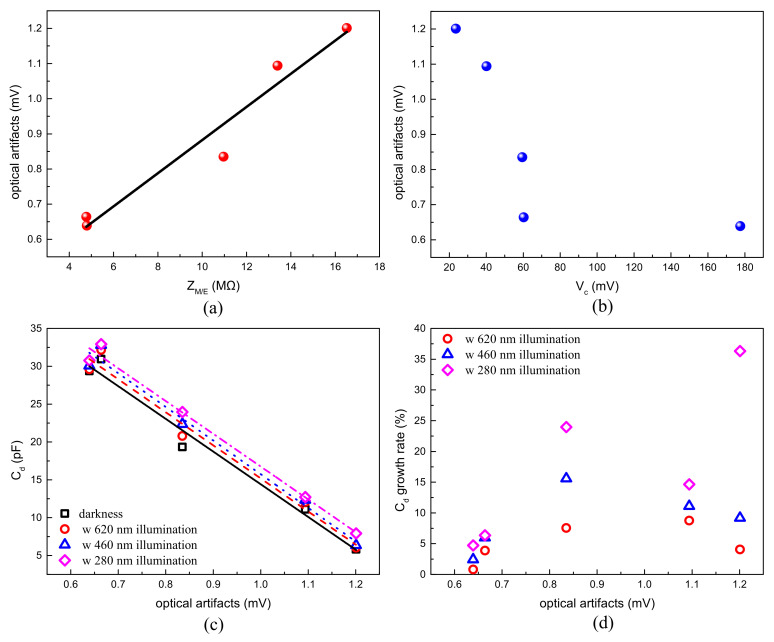
The correlation analyses between optical artifacts and (**a**) Z_M/E_, (**b**) the critical value V_c_, (**c**) C_d_ and (**d**) the growth rate of C_d_, respectively.

**Table 1 micromachines-12-01061-t001:** Comparison of the electronegativity and critical value of microelectrodes with different surface materials.

Material		Ti	Cr	Au	Pt	Ir
Electronegativity scales	X_p_	1.5	1.6	2.4	2.2	2.2
	X_a_	1.32	1.56	1.42	1.44	1.55
	X_Y_	1.47	1.67	1.77	1.79	1.83
Critical value (mV)		40.14	23.50	59.53	60.26	177.53

**Table 2 micromachines-12-01061-t002:** Comparison of various parameters of microelectrodes made by different surface materials.

	Material	Cr	Ti	Au	Pt	Ir
Parameter						
Z_M/E_/MΩ (at 1 kHz)		16.5211	13.3976	10.9625	4.7727	4.7996
Critical value/mV		23.50	40.14	59.53	60.26	177.53
Electrical recoverability/MΩ (after redox reaction)		4.1770	1.4597	5.5946	0.6343	1.7192
C_d_/pF	darkness	5.8127	11.0977	19.3344	30.9437	29.3717
	620 nm	6.0492	12.0689	20.7934	32.1484	29.6067
	460 nm	6.3470	12.3297	22.3501	32.7914	30.0818
	280 nm	7.9230	12.7229	23.9677	32.9115	30.7582
C_d_ growth rate/%	620 nm	4.070	8.751	7.546	3.893	0.800
	460 nm	9.193	11.101	15.597	5.971	2.418
	280 nm	36.306	14.644	23.964	6.359	4.720
Peak-to-peak amplitude of optical artifacts/mV		1.201	1.094	0.835	0.664	0.639

**Table 3 micromachines-12-01061-t003:** Pearson correlation coefficients (|r|) of the recorded optical artifacts and other parameters.

Parameter		Pearson Correlation Coefficients|r|
Z_M/E_/MΩ (at 1 kHz)		0.97589
Critical Value/mV		0.72762
Electrical Recoverability/MΩ		0.36151
C_d_/pF	Darkness	0.98977
	620 nm	0.98919
	460 nm	0.99148
	280 nm	0.99455
C_d_ growth rate/%	620 nm	0.51144
	460 nm	0.48589
	280 nm	0.80052

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
