# Peer review of "Influence of the Surface Material and Illumination upon the Performance of a Microelectrode/Electrolyte Interface in Optogenetics"

_micromachines, 2021, doi:10.3390/mi12091061_

Round 1

Reviewer 1 Report

Authors investigated the effect of surface material and illumination upon the performance of the microelectrode/electrolyte interface. For this, authors fabricated an integrated planar optrode with a blue LED and microelectrodes. In addition, authors modelled and verified a charge transfer mechanism on the interface. Authors demonstrated that surface materials had high influence on charge transfer capacity, stability, photostability, and optical artifacts. Iridium was the best material for optrodes, which improves the signal and decreases optical artifacts. This manuscript is well structured and written. This research work has potential application in optogenetics. However, this manuscript can be improved considering the following comments:

1.-Authors should consider more recent references between 2018 and 2021 about microelectrodes in optogenetics.

2.-Authors should add the main advantages and limitations of their integrated planar optrode with a blue LED and microelectrodes.

3.-Authors should include more technical information about the design of the monolithic integrated planar optrode, considering the main dimensions of the different material layers. In addition, authors should consider more detail information about the fabrication process of the monolithic integrated planar optrode, including the description of temperature, pressure, time, materials, and equipment used in this fabrication process.

4.-Line 163-166. The following sentence is very large. This sentence should be improved:

Then the probe was fixed on the customized printed circuit board (PCB) with glue and wire-bonded onto pads of the PCB, so that power could be supplied to drive the implanted mirco-LED and the neural signals recorded by microelectrodes could be extracted conveniently.

5.-Figure 4 should include the label Frequency (kHz) using first capital letter.

6.-Line 140-418. The following sentence is very large. This sentence should be improved:

The According to the correlation analysis, the amplitudes of the recorded optical artifact had been found to be significantly correlated (|r|=0.98977 @ darkness, |r|=0.98919 @ 620 nm, |r|=0.99148 @ 460 nm, |r|=0.99445 @ 280 nm) with EDL capacitor (Cd) under all the testing illumination conditions, and significantly correlated (|r|=0.97589) with the impedance of microelectrode/electrolyte interface (ZM/E), and moderately correlated (|r|=0.72762) with stability for faradaic processes from the critical value (Vc), but lowly correlated with re-coverability after redox reaction (|r|=0.36151) and photostability from Cd growth rate under the illumination conditions of optogentics (|r|=0.48589 @ 460 nm).

7.-Authors should add more discussion of the results of Figure 8 and Tables 1 and 2.

8.-Format of references must be modified. Authors must check the format of references used in Micromachines.

Author Response

Thanks for reviewer's valuable comments. As for the response and revised manuscript,  please see the attachments.

Reviewer 2 Report

The authors present a systematic comparison of 5 different materials for MEA electrodes, with the intent to be used for in vivo optogenetic applications. The study is careful to separate optical effects on electrical performance from device design effects. An understanding of the relative performance of different electrode types during optogenetic manipulation of the brain is of high importance for the current state of the field, where devices should be transitioning from the experimental stage to the pre-market stage.

Though this topic is of great interest and the experiments presented are sound, the manuscript requires revision. Generally, the text should be amended to not imply that studies with cells were carried out in this investigation. It is true that the physical properties investigated have a bearing on the in vivo performance, but since the relative importance of these properties in a complex biological environment is not directly investigated, the current text overstates how transferable the conclusions are. One prominent example of this is the comparison of the transition from a dominant capacitive coupling between the electrode and the electrolyte to a faradaic conductance at different voltages. Since PBS is a relatively simple buffer solution, various biologically relevant, brain-present, and redox active molecules are neglected that are likely to alter the voltage of the transition.

Furthermore, though it is correct to assume that inherent optical properties of electrode materials and device design related performance deficits may be independently fixed, the end user will be critically concerned about the final recording performance. In that sense, an optically induced capacitive signal and an electrically coupled signal from the LED driving current are both critical points for the end user.

A re-directing of the paper to focus on the material in more depth would separate it from the above concerns, while still allowing the authors to suggest its relevance in multiple fields. This would ideally include a confirmation of the hypothesized mechanism of light induced impedance changes in the different materials and a careful consideration of how production may have affected the outcome. For example, the roughness of the different materials is not presented. If the production of one material results in a rougher film than another material, despite the same nominal electrode opening the real surface area (and therefore all capacitive properties) will be different independent of the atomic electrical properties of the metal. The observations in each of the tested materials is available in the literature and this manuscript rather simplifies the comparison between materials by assuring all of the parameters are consistent. However, without confirming the mechanism of the effects observed, the manuscript's novelty is substantially reduced.

Specific comments to the current version of the manuscript as indicated by line or item number are below.

Please check the correct prepositions are used throughout the manuscript.

Line 98-99 'which have yet to reach an agreement' OR 'which still have to reach an agreement'?

Line 162 Encapsulation and assembly followed the fabrication process.

Line 165 In the current form of the paper ‘implanted’ may lead to misunderstanding meaning ‘in-brain’. 'Embedded' or a similar phrase would make it more clear this refers to the LED's position on or in the optrode.

Line 261-271 Literature has well established that electrode capacitance dominates the interaction with the electrolyte during neural recordings. What was the counter-evidence to motivate the study in this part? If the goal was just to show that C > R and therefore faradaic parameters can be neglected, then the text should be modified to make that point clear.

Line 327-331 Reference 35 seems to not account for more recent use of microelectrode arrays in non-human primates that extend for many more weeks to years. To maintain the 18 week assertion, please describe exactly what comparison is being made and in which animal.

Line 351-364 The steady illumination experiments are good for the theory, but are not related to how the device will be used in the intended application. Only the step function opsins work with long periods of illumination. In most optogenetic tools short flashes are applied, making the initial and final peaks of the artifact the most relevant components.

Line 384 What is a ‘short’ duration relative to the intended signals to be recorded?

Fig 7. Artifacts are 500-900µV. Neuronal signals coupled to a chip typically range from 5µV in vivo to hundreds of µV in vitro, with only exceptional cases reaching over 1mV. How will this affect use of the device?

Fig 8b, What suggests a straight line should be fit through those points?

Line 457 the common usage of ‘injectable biological device’ in this field refers to a device that passes through a needle or uses another shuttle to enter the tissue, as in doi: 10.1126/science.1232437. The proposed device rather seems to claim that the sapphire substrate would allow direct implantation – though data is not shown for this. Please clarify the insertion method and the definitive claims for the sapphire substrate.

Line 470 Biocompatibility was assumed via cited prior art. There is no biocompatibility test described in this evaluation system. Stability and recoverability need to be defined. It is not described how this system tests the ability to recover a device from the tissue where it is implanted, nor what implant relevant times were tested for stability. The performance given in line 327-331 references a group that used a very different system to evaluate devices.

Author Response

Thanks for reviewer's valuable comments. Please see the attachment for the response.

Round 2

Reviewer 1 Report

Authors have improved the second version of their manuscript. This manuscript included more recent references about microelectrodes in optogenetics and the main advantages and limitations of the integrated planar optrode with a blue LED and microelectrodes. In addition, authors added more technical information about the design of the monolithic integrated planar optrode, considering the main dimensions of the different material layers. Also, authors considered more detail information about the fabrication process of the monolithic integrated planar optrode. Authors improved the discussion about the results in Figure 8  and Tables 2 and 3.

Reviewer 2 Report

Thank you for the revisions. In particular, the additional data on the electrode surface shows that Ir is even better than was previously suggested.